# Recent Trends in the Preparation of Nano-Starch Particles

**DOI:** 10.3390/molecules27175497

**Published:** 2022-08-26

**Authors:** Nora Ali Hassan, Osama M. Darwesh, Sayed Saad Smuda, Ammar B. Altemimi, Aijun Hu, Francesco Cacciola, Imane Haoujar, Tarek Gamal Abedelmaksoud

**Affiliations:** 1Food Science Department, Faculty of Agriculture, Cairo University, Giza 12613, Egypt; 2Agricultural Microbiology Department, National Research Centre, Dokki, Cairo 12622, Egypt; 3Department of Food Science, College of Agriculture, University of Basrah, Basrah 61004, Iraq; 4College of Medicine, University of Warith Al-Anbiyaa, Karbala 56001, Iraq; 5College of Food Science and Engineering, Tianjin University of Science & Technology, Tianjin 300457, China; 6Department of Biomedical, Dental, Morphological and Functional Imaging Sciences, University of Messina, 98125 Messina, Italy; 7Laboratory of Biotechnology and Applied Microbiology, Department of Biology, Faculty of Sciences of Tetouan, Abdelmalek Essaadi University, Tetouan 93000, Morocco

**Keywords:** starch nanoparticles, morphology, SNPs methods, SNPs functional properties

## Abstract

Starch is affected by several limitations, e.g., retro-gradation, high viscosity even at low concentrations, handling issues, poor freeze–thaw stability, low process tolerance, and gel opacity. In this context, physical, chemical, and enzymatic methods have been investigated for addressing such limitations or adding new attributes. Thus, the creation of biomaterial-based nanoparticles has sparked curiosity. Because of that, single nucleotide polymorphisms are gaining a lot of interest in food packaging technology. This is due to their ability to increase the mechanical and water vapor resistance of the matrix, as well as hide its re-crystallization during storage in high-humidity atmospheres and enhance the mechanical properties of films when binding in paper machines and paper coating. In medicine, single nucleotide polymorphisms (SNPs) are suitable as carriers in the field of drug delivery for immobilized bioactive or therapeutic agents, as well as wastewater treatments as an alternative to expensive activated carbons. Starch nanoparticle preparations can be performed by hydrolysis via acid hydrolysis of the amorphous part of a starch molecule, the use of enzymes such as pullulanase or isoamylase, or a combination of two regeneration and mechanical treatments with the employment of extrusion, irradiation, ultrasound, or precipitation. The possibility of obtaining cheap and easy-to-use methods for starch and starch derivative nanoparticles is of fundamental importance. Nano-precipitation and ultra-sonication are rather simple and reliable methods for nanoparticle production. The process involves the addition of a diluted starch solution into a non-solvent, and ultra-sonication aims to reduce the size by breaking the covalent bonds in polymeric material due to intense shear forces or mechanical effects associated with the collapsing of micro-bubbles by sound waves. The current study focuses on starch nanoparticle manufacturing, characterization, and emerging applications.

## 1. Introduction

Many plants generate starch as a source of stored energy in the form of polysaccharides. Starch is a natural, renewable, and biodegradable polymer. It is the second most abundant biomass material and is a carbohydrate storage product found in all chlorophyll-containing plants such as corn, potato, rice, wheat, and barley. If it is extracted from a plant, it is referred to as “native starch,” whereas undergoing one or more modifications (enzymatic, chemical, or physical) to achieve specific properties categorizes it as “modified starch” [1]. Starch is made up of two major components: Amylose (a linear or slightly branched (1 4)-D-glucan) and amylopectin, a highly branched (1 4)-D-glucan (Figure 1).

Starch is derived from many crops and is categorized according to its botanical source, which includes cereals, legumes, vegetables, and roots/tubers. Cereal starches derived from maize, whose scientific name is *Zea mays* L., are one of the primary food crops and a highly unique plant. Maize starch accounts for approximately 80% of the world’s starch production, and it is extracted from corn pieces (content 64–80%) using a wet method. Corn starch is utilized in a wide range of meals and uses. Basic maize starches contain traces of protein (0.35 percent), fat (0.8 percent), ash, and more than 98 percent of two polysaccharides, amylose and amylopectin (carbohydrates). Starch is derived from plant sources that are insoluble in water and is in the form of granules at room temperature [2].

Starch granules vary in size and form, with the majority falling in the range of 2–100 mm, with maize’s size ranging between 2 and 30 µm and its shape round and polyhedral. Whereas lentil has a size ranging between 6–37 and 6–32 µm and their shape is oval, spherical, and elliptical, sweet potato’s size ranges between 1 and 100 µm and has shapes that are oval, spherical, and round. Moreover, tapioca’s size ranges between 5 and 45 µm and its shape is spherical/lenticular. Furthermore, rice’s size is less than 20 µm and ranges between 2 and 8 µm, while its shape is polygonal and angular. In wheat, the size ranges between 10 and 30 µm and is disc-shaped. Faba bean has an irregular or oval shape and the size ranges between 20 and 48 µm [3]. The search for literature on the Web of Science was conducted using the keywords “Nano-Starch” or “Starch”, and 136 research and review articles were identified for this review. We include a detailed overview of the knowledge gained over the years in this review article, including recent trends in the preparation of nano-starch particles, and future research needs.

## 2. Starch Nanoparticles

Nanoparticles are solid or colloidal particles with sizes ranging from 1 to 100 nm. The size of the particles influences the chemical reactivity, flowability, opacity, stability, and material strength of many materials [4]. Nanoparticles are classified into three types based on their nature: Organic nanoparticles, inorganic nanoparticles, and mixed organic/inorganic or surface-modified nanoparticles. Metals or metal oxides such as gold, silver, zinc oxide, titanium dioxide, synthetic amorphous silica, aluminum oxides, calcium carbonate, cerium dioxide hydroxides, and carbon-based compounds such as carbon are used to make inorganic nanoparticles.

There has recently been renewed interest in the creation of biomaterial-based nanoparticles. According to research, a wide range of organic and edible nanoparticles may be made from food-based components such as polysaccharides, lipids, proteins, minerals, and surfactants [5]. The most common model for starch is a concentric semi-crystalline multi-scale structure that allows for the creation of new nano elements: (i) Starch nanocrystals formed by acid hydrolysis of amorphous domains in semi-crystalline granules and (ii) starch nanoparticles formed from gelatinized starch [6]. So, starch is a natural substance or a nanoparticle. The advantages and drawbacks of the preparation of starch as nanoparticles (SNPs) are shown in Figure 2 [7,8,9,10].

## 3. The Classification Basis of the Preparation Method of Starch Nanoparticles

The preparation methods of starch nanoparticles (SNPs) can be classified into three categories: (1) Hydrolysis (acid and/or enzymatic hydrolysis, or a mix of the two); (2) renewal/micro perception; and (3) physical methods (milling, high-pressure homogenization, electrospraying, gamma radiation, and ultra-sonication).

### 3.1. Acid Hydrolysis

Acid hydrolysis has been used for years to prepare SNPs using mild acids and temperatures below starch’s gelatinization threshold. Mineral acids including oxalic acid, sulfuric acid, and hydrochloric acid are employed, although citric acid has also been used, at temperatures ranging from 25 to 55 °C [11]. The acid hydrolysis of starch nanoparticles was performed with five factors in mind: Temperature, acid concentration, starch concentration, hydrolysis duration, and agitation [12]. Acids cause the glucosidic linkages to break, changing the structure and properties of the native starch. Acids dissolve only the amorphous region, leaving the crystalline region intact [13]. Acid hydrolysis is broken down into three steps: (i) Surface erosion; (ii) collapse of the granule wall owing to acid radial diffusion; and (iii) fragmentation of the growth rings [14]. Two explanations have been proposed to explain why hydrolysis is so much slower. To begin with, the compact packing of starch crystallites with starch chains inhibits H_3_O^+1^ from easily penetrating the areas. Second, acid hydrolysis of a glucosidic link may cause the D-glucopyranosyl portion to change shape (from chair to half chair). If the sugar conformation is inhibited by the crystalline structure, this transition will be impossible. Winarti et al. [15] conducted a study of acid type and duration of hydrolysis on SNPs generated from taro and arrowroot starches. The study concluded that utilizing HCL instead of H_2_SO_4_ in hydrolysis resulted in a higher yield and larger particle size. In contrast to HCL, which increased the particle size by increasing the hydrolysis duration, the data showed that a high H_2_SO_4_ hydrolysis duration decreases particle size. Furthermore, they stated that the yield of SNPs had dropped as a result of the protracted hydrolysis process, which caused excess starch components to dissolve in the acidic medium, lowering the yield of SNPs. Wang et al. [16] treated maize starch with three concentrations of HCL: 0.06, 0.14, and 1.0 N at 50 °C. They stated that increasing the acid concentration increased the rate of hydrolysis and crystallinity. Velásquez et al. [17] stated that increasing the hydrolysis temperature had no impact on the crystallinity index of quinoa SNPs, but that it did reduce the yield, thermal stability, and particle size of SNPs. Kumari et al. [18] reported that the acid hydrolysis method was used to produce mung bean SNPs. The starch dispersion was agitated at 100 rpm for seven days at 37 °C in an aqueous sulfuric acid solution. The hydrolyzed starch suspension was centrifuged after acid hydrolysis. Following acid hydrolysis, irregular or round-shaped nano-scale crystals with an average hydrodynamic diameter of 179 nm were produced. Mung bean SNPs had a lower negative zeta potential than their native starch. Figure 3 shows acid hydrolysis for the preparation of starch nanoparticles.

### 3.2. Nano-Precipitation

Two miscible solvents are used to manufacture nanoparticles via the nanoprecipitation method: One as a good solvent (typically an organic solvent such as ethanol, isopropanol, or acetone) and the other as a non-solvent for the chemical that will form the particle (i.e., lipid, polymer). The requirements for the nanoprecipitation procedure are the preparation of an organic phase and a non-solvent phase, commonly referred to as the aqueous phase, both of which ensure the entire solubility of all initiating components. SNPs have been made from a variety of botanical sources, including maize, tapioca, sweet potato, potato, and pea starch, by adding ethanol drop-wise to a weak solution of gelatinized starch while stirring. Several factors influence the properties of SNPs, including the starch content, surfactant addition, and the ratio of starch solution to non-solvent employed in the precipitation. Surfactants boost the crystallinity of SNPs according to researchers, which is due to the surfactant’s capacity to infiltrate the starch interior cavity of the single-helix portion. Jiang et al. [19] conducted a study wherein quinoa starch nanoparticles (QSNPs) were made by the nano-precipitation method, which was modified from a previously published process. For the dissolving of natural quinoa starch, a dimethyl sulfoxide (DMSO)/H_2_O solution was utilized as the solvent system. The gelatinized starch solution was chilled before being dropped into various amounts of 100% ethanol. QSNPs had a lower particle size of 166.25 nm and a 26.62% higher loading capacity. Through hydrogen bonding, starch nanoparticles (SNPs) interacted with quercetin. Because of their tiny particle sizes and high loading capacities, QSNPs were more successful in protecting and prolonging quercetin bioactivity. Nanoparticles are spontaneously formed when the organic phase is dropped or added in a one-shot manner to the aqueous phase [20]. The benefits include more homogenous nano-metric particles and reduced usage of hazardous solvents. When compared to other synthesis techniques, nanoprecipitation is a more promising technology since it is a simple process with a low cost and less danger of sample contamination [21]. It has been extensively utilized to prepare SNPs in the pharmaceutical industry [22]. It involves gradually adding a diluted polymer solution to a non-solvent or sequentially adding the non-solvent to the polymer solution, resulting in the production of nano-scale polymer particles [23]. The process is mainly based on the deposition of interfacial biopolymers and the displacement of a lipophilic solution by a water-miscible semi-polar solvent. SNPs were made from a variety of botanical sources, including maize, tapioca, sweet potato, potato, and pea starch, by adding ethanol drop-wise to a weak solution of gelatinized starch while stirring. SNPs had a particle size of 30–75 nm [24]. Several factors influence the properties of SNPs, including starch content, surfactant addition, and the ratio of a starch solution to a non-solvent employed in the precipitation. The effects of ionic and non-ionic surfactants (Cetyl tri-methyl ammonium bromide and Sodium dodecyl sulphate) on the crystalline structure of SNPs made from potato starch have been investigated. Surfactants boost the crystallinity of SNPs according to researchers, which is due to the surfactants’ capacity to infiltrate the starch interior cavity of the single-helix portion. The study stated that ionic surfactants provided better drug absorption than non-ionic surfactants, which was explained by the fact that ionic surfactant charges boost surfactant molecules’ attraction to starch particles [25]. Figure 4 shows the nano-precipitation method of SNPs.

### 3.3. Milling

Milling is an environmentally friendly method of producing starch nanoparticles that may directly influence the physicochemical properties of starches, resulting in more amorphous nanoparticles. Lu et al. [26] investigated the influence of process factors in the synthesis of maize nanoparticles by stirred-medium milling (266.4–309.8 nm and 700 nm). Stirred-media milling is a time- and energy-intensive method for producing nanoparticles. Furthermore, product contamination from grinding media wear and mill surfaces limits the use of stirred media milling technologies in many applications requiring extremely high purity [27]. A shaft with a series of impellers is located inside the milling chamber, providing high-shear agitation (up to 4000 rpm). A 600 mL chamber, for example, can be linked to a vessel of 5 to 10 L and loaded with the milling medium and suspension. The suspension then enters the milling chamber, where it is subjected to strong media grinding before exiting through a tiny aperture. The total milling time depends on how long the material is in the chamber. In terms of suspension characteristics, milling time is usually reduced as the suspension viscosity increases owing to increased shear pressures and energy input, as well as when drug loading increases due to higher particle densities in the grinding zone [28]. Ball milling has been shown to alter the shape, crystallinity, and molecular weight of starch granules from various sources [29]. The gap causes the milling medium to be stretched and held within the mill. The mill and the vessel are both jacketed to keep the temperature under control [30]. Ahmad et al. [10] collected samples and stored them at −18 °C in polyethylene zip pouches, which resulted in the starch nanoparticles being made by planetary ball milling from three underutilized and new sources of starch: Horse chestnut, water chestnut, and lotus stem. Water chestnut and horse chestnut starch nanoparticles created smaller, more stable particles than lotus stem starch nanoparticles, according to the study. Furthermore, the produced starch nanoparticles were amorphous, with higher viscosity and heat stability than native starch, and could be used in a variety of food processing applications. Ball milling can effectively manufacture starch nanoparticles, according to the study, and may offer some benefits over chemically synthesized nanoparticles in terms of safety, yield, and cost-effectiveness.

### 3.4. Gamma Radiation

This technique can break big molecules into tiny pieces and cleave glycosidic-bonds, therefore gamma radiation can be employed to generate starch nanoparticles. The method involves stirring starch and boiling water to make a homogeneous paste, then irradiating the suspension with gamma rays, which produces active free radicals that cause the starch to hydrolyze. Cleavage occurs in the amorphous portions of starch rather than the crystallite regions, resulting in fragmentation. The gamma radiation approach is remarkably similar to the starch acid hydrolysis method in this regard. The nanoparticles produced by this process typically have a diameter of less than 100 nm. Furthermore, due to the huge number of OH groups on their surface, these nanoparticles contain nano-crystal aggregates, which become firmly connected by hydrogen bonding, resulting in rapid thermal deterioration [31]. Bashir and Haripriya [32] reported that the swelling capacity of gamma-radiated wholewheat starch samples with doses of 0.5, 1, 2.5, 5, and 10 kGy increased dramatically. It was determined that starch swelling was caused by the breakdown of the amylopectin chain. The restriction of water entering the starch matrix following gelatinization resulted in a decrease in swelling property at 90 °C. Verma et al. [33], Wani et al. [34], and Sunder et al. [35] determined that syneresis is another property of starch that is important to consider while evaluating the modification process. Syneresis is a phenomenon in which gelatinized starch removes extra water content, which is a bad thing. Syneresis study of gamma-irradiated horse chestnut starch revealed a consistent decrease in starch syneresis only when the dosage was increased. Due to crystallization, amylose deformation, and amylopectin aggregation occurring later in the storage time, 15 kGy gamma-treated starch showed the lowest rate of syneresis after 120 h.

### 3.5. Electro Spraying

Electro-spraying is a novel method for creating nanoparticles from various biopolymers (starch, alginate, gelatin, chitosan, agar, etc.). This method is based on the premise that an electric field bends droplets in the micrometer and nanometer range. A Coulomb force is created in the droplets by applying an electric field, which simulates cohesive forces and Coulomb forces. The cohesive forces release surface tension, resulting in the production of nano-droplets. The electro-spray principle is based on the ability of an electric field to distort the droplet interface and produce droplets in the micro-meter or nano-scale range, depending on the parameters regulated. When the Coulomb force is applied, it is capable of overcoming the droplet’s cohesive force, which manifests in surface tension, allowing it to be free to create nano-droplets. These variables include system characteristics such as the solution flow rate, applied electric potential, collector distance, solution viscosity, conductivity, surface tension, molecular weight, and polymer concentration [36]. Then, electrostatic spraying combined with ethanol precipitation was used in this study to produce debranched starch nanoparticles (DSNPs). The starch nanoparticles were between 243 and 337 nm in size, and the DSNPs had good dispersibility. The distance between the needle tip and the collector was kept at 15 cm during the electrostatic spray process, the working chamber temperature was kept at 55 °C, and the speed of delivery was kept at 0.05 mm/s. The sprayed material was incubated in the petri dish for 4 h before being centrifuged at 12,000× *g* for 15 min. The obtained precipitate was freeze-dried for three days at 60 °C [37].

### 3.6. High-Pressure Homogenization

The chemical, pharmaceutical, food, and biotechnology sectors all use high-pressure homogenization. Changes in the products, as well as the particles, colloids, or macromolecules that make up the products, may occur throughout this therapy. The starch slurry is passed through a micro-fluidizer, which can be intensified by external pressure sources, external mechanical pumps, integrated mechanical micro-pumps, or electro-kinetic mechanisms, resulting in the breakage of the starch slurry. The persistent pressure of 207 MPa was maintained. The starch PS was reduced from 3–6 µm to 10–20 nm without any change in crystal structure or thermal stability of starch granules [38]. The pressure applied, the starch concentration and source, and the treatment period all affect how starches react to HPH. Changes in starch quality are caused by a variety of pressure-induced processes [39]. Thus, innovative applications for the high-pressure homogenization of starch nanoparticles are being investigated. The treated samples were freeze-dried for approximately 24 h at 80 °C with the condenser temperature, then sieved through a 100-mesh sieve, sealed, and stored at room temperature in a desiccator. During the HPH treatment, the LS absorbed a great deal of water and quickly inflated, increasing the size distribution of compressed LS and boosting starch particle (SP) and solubility. Furthermore, Apostolidis and Mandala [40] reported that the HPH treatment was successful in reducing the particle size of high amylose maize starch from approximately 7.5 m to 0.5 m. As a result, non-thermal mechanical treatment for starch particle size reduction was found to be effective. Damage effects of starch, but also amylo-pectin structure disruption, were seen at 250 MPa. There were also relationships between particle sizes, crystalline structure, and amylose content. Furthermore, Chutia and Mahanta [41] employed the optimum ultra-sounded-treated starch suspension, which was homogenized directly in a high-pressure homogenizer under optimal nanoemulsion preparation pressure conditions. The lyophilized starch suspension was homogenized. The optimum high-pressure homogenized starch suspension was coded after it was freeze-dried.

### 3.7. Ultrasonication with/without Acid Hydrolysis

The addition of a solution in which the starch is diluted into an aqueous phase in the presence of ultrasonic homogenization reduces the size of the particles formed as a result of shear forces or mechanical effects produced by the collapse of microbubbles that form sound wafers [42,43,44]. This technique, however, has been used in conjunction with acid hydrolysis because the combination of physical and chemical processes creates nanoparticles with more desirable characteristics [45]. This method is easy and convenient in terms of safety and cost, and it may provide a higher yield with the appropriate particle size. Ultrasonication is exposed to numerous parameters such as the frequency, power, time, temperature, source, and type of starch dispersions, which are expected to have a wide range of sizes, ranging from 30 nm to 200 nm [46].

Ultrasonic atomization is a potential method for producing liquid micro-droplets using ultrasonic radiation. Two primary theories have been proposed in the droplet formation mechanism to explain the thin vibrating liquid film breakdown during ultrasonic atomization: Capillary waves and the cavitation mechanism [47]. Capillary waves are made up of an ordered mesh-like structure that has the same number of ridges and valleys per unit area [48]. Droplets detach from the crests of stationary capillary waves in the conventional model of ultrasonic atomization, and cavitation bubbles develop on the vibrating capillary waves in the cavitation process. The ultrasonic atomization of bovine serum albumin-based nanopowder for curcumin encapsulation is performed in a single step, employing an ultrasonic piezoelectric oscillator, which produces nanoparticles with a size of 200–800 nm and a droplet size of less than 1 m, as well as spherical nanoparticles with a uniform smooth surface and particle sizes in the ranges of 200–800 nm. Due to the relative ease, low price, and ease of gathering the formed powder, this approach has advantages over others. The variety of preparation conditions provides for the development of more complex structures with greater control over their composition and size. The method can be used with a wide range of low-viscose biopolymers, decreasing the production time and cost greatly [49]. Figure 5 shows ultrasonication with/without acid hydrolysis, while Figure 6 shows ultrasonic atomization method.

### 3.8. Enzymatic De-Branching with/without Acid Hydrolysis

Other options for acid hydrolysis include the use of enzymes to de-branch starch chains, which has been performed with enzymes such as Beta-amylase, Alpha-amylase, glucoamylase, and pullulanase. Alpha-amylase is an endo-amylase that causes random cleavage of internal Alpha-1, 4 bonds, whereas Beta-amylase is an exo-amylase that causes cleavage of alternating Alpha-1, 4 bonds from non-reducing ends, which is referred to as hydrolysis rather than de-branching [50,51] Pullulnase and isoamylase are carbohydrases that hydrolyze α-1,6 glycosidic linkages into branched-chain amylopectin, which is a promising method after the α-1,6 links that originated are removed. The linear pieces of starch generated by de-branching enzymes re-associate during starch retro gradation, creating more crystalline structures. This recrystallization of linear glucans following the de-branching of native starch has also been utilized to produce SNPs. The crystallinity of the nanoparticles was enhanced by recrystallization. However, when the recrystallization time increased (48 and 72 h), the number of crystals obtained decreased. However, such crystallites were less robust than starch nanocrystals (SNCs) produced by acid hydrolysis, and it was determined that enzymatic pretreatment of starch combined with acid hydrolysis reduces the time from days to hours while increasing the yield. Although it saves time, the expense is rather high. In contrast, only enzymatic hydrolysis offers an advantage over acid hydrolysis in terms of speed and yield (approximately 55%). This technique has an advantage over acid hydrolysis in that no chemical reagents are employed; therefore, no washing is required prior to freeze-drying [52]. Lin et al. [53] reported that Lotus seed starch nanoparticles were produced by ultrasonication in different powers, times, and liquid ratios (starch: buffer solution) with the addition of enzymatic hydrolysis. Regarding the structure and physico-chemical properties of lotus seed starch nanoparticles produced by ultrasonic-assisted enzymatic hydrolysis without ultrasonic treatment, the particle size distribution of lotus seed starch nanoparticles is 16.7–2420 nm. Ultrasonic-assisted enzymatic hydrolysis degraded lotus seed starch nanoparticles, reducing their size. Ultrasonic power and time increased starch nanoparticles’ size and crystallinity in the treatment group.

## 4. Comparison of Production Rate of Starch Nanoparticles Preparation Methods

Acid hydrolysis is a straightforward and easy-to-control method of producing nanoparticles. However, selective hydrolysis is a rather slow process that results in a poor yield. Another disadvantage is the high concentration of hydroxyl groups, which form aggregates due to the supramolecular interactions of starches. This propensity makes SNPs unsuitable for industrial use. Enzymatic hydrolysis is a non-hazardous and ecologically acceptable technique. The regeneration settings were adjusted to regulate the size and form of the SNPs. The process, however, is not complete when the enzyme activity drops. In the food industry, enzymatic hydrolysis of starch nanoparticles is frequently employed. In comparison to acid hydrolysis, ultrasonic treatment is faster, easier to handle, and does not need several rinses because no solvent is employed. The drawbacks of this approach are crystalline structure distortion and, depending on the angle of X-ray diffraction, amorphous or poor crystallinity structure. The applications of SNPs produced using ultrasonic techniques are quite diverse. Depending on the use, the initial starch should be of varying purity. The combination of hydrolysis and ultrasonic treatment resulted in a higher yield (78%) as compared to acid hydrolysis alone, which generated just 15%. The drawbacks of the technique are that SNPs were not sufficiently stable and inflated readily when sonicated, leading to starch aggregation formation [54]. To employ the current hydrolysis and ultrasonic method for generating SNPs, more precise control over the hydrolysis and ultra-sonication is required. The combination of enzymatic hydrolysis and the recrystallization technique has a high yield (approximately 55%), and no chemicals are introduced throughout the synthesis. The drawback of this technique is that the starch nanoparticles vary in size. This hydrolysis and ultra-sonication technique also has the benefit of not requiring the addition of any chemical reagents during the preparation phase. This method of emulsion crosslinking avoids the use of hazardous chemicals (such as cyclohexane) and is both efficient and ecologically benign [54]. Chen et al. [55] employed the acid hydrolysis technique to make corn starch nanoparticles with the morphology of nanoplatelets and a size of 107 nm. Cuthbert et al. [56] used an enzymolysis process to manufacture maize starch nanoparticles with irregular shapes and sizes ranging from 2.4 to 6.7 nm. In contrast, Bel Haaj et al. [57] utilized ultrasonication to generate maize starch nanoparticles and obtained a granular and platelet shape with sizes ranging from 30 to 100 nm. However, when acid hydrolysis and ultrasonic techniques for maize starch nanoparticles are combined, globular forms ranging from 50 to 90 nm are formed. Kim et al. [58] indicated that other procedures, such as homogenization, which produced pores particles with a size of 540 nm, represented this form of starch in large sizes, in contrast to other ways. Apostolidis and Mandala [40] used milling and produced gel-like particles with a diameter of 245 nm [59]. Despite this, Hebeish et al. [60] used a nano-precipitation method to simulate maize starch with sizes ranging from 135 to 155 nm in spherical forms. Lately, Harsanto et al. [61] investigated a nano-precipitation technique to simulate breadfruit starch with sizes less than 200 nm with a uniform size distribution, porous surface, and an amorphous phase. Finally, Table 1 shows the comparison of the production rates of starch nanoparticle preparation methods.

## 5. Techniques for Characterization of SNPs Properties

### 5.1. Thermal Properties

#### 5.1.1. Differential Scanning Calorimetry

Differential scanning calorimetry (DSC) is a thermal analysis technique that determines how much energy a sample absorbs or emits as a function of temperature. A DSC equipment diagram, entropy, enthalpy, and special heat determination may all be used to determine the temperature and quantity of heat flow in a sample [65]. Liu et al. [66] assessed the temperature and gelatinization enthalpy by conducting experiments that were carried out with a nitrogen gas flow rate of 30 mL/min and an empty pan for reference. To eliminate the moisture, the crushed sample was dried in a laboratory oven (105 °C) for 6 h. The aluminum pan was filled with dried powder samples (3–5 mg) and 6–10 L of water. The pans were sealed and heated at a rate of 10 degrees Celsius per minute, with temperatures ranging from 50 to 125 °C. The onset (To), peak (Tp), conclusion temperature (Tc), and gelatinization enthalpy were all measured using DSC thermo-grams and that resulted in SNPs having much higher onset, peak, and conclusion temperatures (90.38–92.08 °C, 105.83–107.57 °C, and 114.53–117.49 °C, respectively) than the native starch (65.59 °C, 72.12 °C, and 79.57 °C). The higher the degree of crystallite perfection and stability, the higher the gelatinization temperature. The To, Tp, and Tc of SNPs constructed with different durations did not appear to change significantly. Despite this, Chacon et al. [67] reported that the absence of an endothermic peak associated with the gelatinization of starch in SNPs indicates that this nanomaterial is an amorphous modified starch.

#### 5.1.2. Thermogravimetric Analysis

Thermo-gravimetric analysis (TGA) is a thermal analysis technique that measures the mass of a sample over time as the temperature varies. It is used to assess a sample’s thermal stability. This study involves knowledge of physical, but also chemical, phenomena. The temperature or time curve, also known as the thermogravimetric curve, displays data on the change in mass with the temperature/time of the sample using TGA as a graph/curve. A derivative plot of the TGA curve, termed DTG, displays the rate of mass change and the rate of mass loss against the temperature curve. Changes in sample mass can occur as a result of processes such as evaporation, dryness, desorption or adsorption, sublimation, and thermal breakdown. These mass shifts are evidenced by phase alterations in the TGA curve or DTG curve peaks [68]. Shapi’I et al. [69] demonstrated that (TGA) was carried out in a nitrogen atmosphere using a thermogravimetric analyzer. TGA analysis was used to assess the thermal stability of films without (chitosan nanoparticles) CNPs, with a low concentration of CNPs, and with a high concentration of CNP on neat starch, 15 percent *w*/*w* starch/CNP, and 30 percent *w*/*w* starch/CNP films. TGA analysis was carried out on a 20 mg sample film at a rate of 10 °C/min in the temperature range of 25–600 °C. TGA determines how much weight each sample loses throughout a temperature range of 25 to 600 °C.

#### 5.1.3. Differential Thermal Analysis

Differential thermal analysis (DTA) is a technique quite similar to differential calorimetry scanning thermoanalytics. In DTA, the analyte undergoes thermal cycles, thus all variations during the analysis are recorded [70,71]. Gardouh et al. [72] used thermal analysis of native starch and SNPs was carried out using an instrument calibrated by indium, and samples (1–1.5 mg) were precisely measured. Then it was put in an aluminum pan and heated at a rate of 10 °C/min from 25 to 300 °C with nitrogen as a purging gas at a flow rate of 15 mL/min. The gelatinization temperature was ascribed to one endothermic peak on the thermogram of natural starch, which was approximately 50 °C. The SNP thermogram revealed two endothermic peaks at 40 °C and 278 °C. The pasting temperature of SNPs is indicated by the peak appearing at approximately 40 °C in the SNP thermogram. However, the amorphous nature of SNPs and the overlapping of endothermic processes at the gelatin temperature caused a broad glass transition. The pasting temperature was found to be substantially lower than that of native starch, demonstrating that SNPs were easier to gelatinize than native starch. This can be interpreted as a difference in helix structure, with the single helix of SNPs being more prone to destruction than the native starch’s double-helix structure.

### 5.2. Mechanical Properties

#### 5.2.1. Dynamic Mechanical Analysis

Dynamic mechanical analysis (DMA) is a method for studying the kinetic characteristics of a sample by estimating strain or stress. It is used to measure polymer materials of various kinds utilizing various deformation modes. There is tension, compression, dual bending of cantiles, three-point bending, and shear modes, and the optimum type should be chosen based on the sample form, modulus, and measurement purpose [70]. Ochigbo et al. [73] reported that on a Perkin Elmer Diamond DMA in-stretching mode, DMA curves of pure Total Plastic Starch (TPS) and TPS nano-composites (with 1 and 3 wt % inorganic content) conditioned at 35, 57, and 75%. RHs were recorded in the temperature range of 120 to 150 °C. Analyses were carried out at a frequency of 1 Hz and a temperature of 2 °C/min. An increase in RH reduces the storage modulus of the films in the temperature range of 75 to 50 °C. The sample conditioned at % RH, on the other hand, reduced faster at higher temperatures (approximately 50 °C). The “lower” and “upper” relaxation transitions, at 40 °C, could be seen in the temperatures and curves at 50 and 100 °C, and they are usually referred to as the “lower” and “upper” transitions. These two processes result from the cooperative interactions of chain segments related to the glycerol–starch and pure-starch-rich domains, owing to the extremely heterogeneous composition of the pure TPS matrix. Small moisture losses during testing above 0 °C can cause dramatic changes in visco-elastic characteristics. These two processes are the outcome of the pure TPS matrix’s extremely heterogeneous nature, i.e., they arise from the coordinated interactions of chain segments related to the glycerol–starch and pure-starch-rich domains. With an increase in RH, the lower transition, which is connected to the glass transition of the glycerol–starch mixture, moves toward lower temperatures. Additionally, the dynamic modulus, which is typically correlated with the storage modulus, can be used to measure DMA in storage (E0). It frequently refers to a material’s “motion” and establishes whether a sample is stiff or flexible. E0 is seen as a material characteristic or the ability to hold onto the energy provided to it. The loss modulus (E00), also known as the dynamic loss modulus, is related to “internal friction”. It is sensitive to various molecular movements, transitions, relaxation processes, morphology, and other structural heterogeneities [74].

#### 5.2.2. Zeta Potential Analysis

Parupudi et al. [75] investigated determined both the scientific community and regulatory organizations regard zeta potential as a metric that determines surface charge. It is a determination of the electrostatic potential difference between a nanoparticle and the bulk solution. A laser is used to pass through a nanoparticle solution while being affected by a variable electric field in order to identify the zeta potential. It is also related to the frequency shift or phase shift of the light scatter produced by the Doppler effect when assessing highly aggregated samples or measuring nanoparticles in high salt buffer conditions. Another function of zeta potential is used to evaluate the surface load of nanoparticles in solution (colloids). The nanoparticles’ surface load draws a thin layer of oppositely charged ions that are present on the nanoparticles [76]. Along with the nanoparticles, this double layer of ions diffuses into the solution. The electrical potential at the dual layer’s boundary is known as the particle’s zeta potential, and it generally ranges between +100 mV and 100 mV. The magnitude of the zeta potential for colloidal stability predicts colloidal stability. Zeta nanoparticles with values more than +25 mV or less than +25 mV are generally quite stable. Because of Van Der Waals inter-particle interaction, dispersions with low potential value ultimately coalesce. The zeta potential is a useful technique for determining the surface condition and long-term stability of nanoparticles [77]. Tagliapietra et al. [78] reported that when starch nanoparticles are added to emulsions and solutions, the zeta potential is an important element. Because the surface charges of these nanoparticles can be changed by other ingredients during the emulsion production, evaluating the zeta potential of starch without any type of application seems to be a useless parameter. Using dilution with deionized water, Gardouh et al. [72] assessed the use of zeta potential in starch nanoparticles. For the optimal formulation, the electrical charge of SNPs was determined using a Malvern size distribution analyzer at 25 °C. Water has a 1.330 refractive index and 0.89 cp viscosity, which resulted in Zeta potential showing a level of −200.25 mV, indicating good formulation stability. This revealed that the hydroxyl groups present on the surface of SNPs are the source of the negative charge.

### 5.3. Physiochemical Properties

#### 5.3.1. Atomic Force Microscopy

AFM enables three-dimensional simulation and analysis, and individual particles and particle classes may be solved [79]. AFM is one of the most widely used types of microscopy for scanning samples. From an experimental standpoint, nanoparticles are used to evaluate AFM tip changes or nanoparticle manipulation, and the interaction of nanoparticles with the AFM probe has been widely studied. When spherical nanoparticles are put on an optimally flat substratum, the nanoparticle height from the AFM picture may be readily calculated [80]. This number is unaffected by the effects of tip–sample convolution and can produce accurate nanoparticle testing. As a result, the statistical findings of nanoparticles are dependent on the right selection and use of AFM data assessment methods, which introduces human error into the entire measurement process [81]. AFM has many advantages over other microscopic techniques, such as high-resolution capacity, minimal sample preparation (samples are in an almost native condition), compatibility for non-conductive samples, flexible imaging environment (in liquid or gas), and assessment of ultra-small structures [82]. Cuthbert et al. [56] conducted AFM in the tapping mode at room temperature with a silicon cantilever that had a maximum tip diameter of 10 nm. The samples were dissolved in ethanol (0.1 g) (2 mL). The resulting suspensions were vacuum spin-dried on a glass slide, dried in a forced convection oven at 45 °C for 2 h, then scanned at room temperature using 512 lines per sample and a 0.5 Hz scan rate over a 0.5 m scan area. In order to realize and determine artifacts caused by the cantilever’s interaction with the sample during interpretation, height and phase pictures were recorded and showed that the average height of the unique nanoparticles, which were separated and measured using AFM in this study, was identical to that previously reported for starch nanoparticles containing complex stearic acid. Bernardo et al. [83] combined glycerol-plasticized starch films with starch nanoparticles in a single, rapid step for use as biodegradable packaging according to the green chemical principle. So, before the film was made, the process’ nanoparticles were examined. Following decantation, a Pasteur pipette was used to take one sample of the clear solution that was suspended.

#### 5.3.2. Transmission Electron Microscopy

Transmission electron microscopy (TEM) technology uses an electron beam to image a nanoparticle sample, providing far better resolution than light-based imaging methods [84]. The best method for precisely calculating the thickness, grain composition, size, and shape of nanoparticles is to use TEM. Recently, TEM has been utilized to determine the impact of nano-composite materials on biological systems [85]. Liu et al. [86] investigated a transmission electron microscope to examine the shape and size of SNPs, and the SNPs that were loaded with peppermint oil (PO) at an acceleration voltage of 80 kV were used to prepare dry samples for further study, and a droplet of SNP or PO-loaded SNP suspension was diluted (5:1000) with distilled water and drop-cast onto a carbon-coated copper grid (400 meshes). This procedure took more than 6 h and resulted in different chain lengths of short linear glucan (SLG). In the present study, ethanol was added to the primary (short linear glucan) aqueous solution. As a result, short SLG (S-SLG) was retained in the supernatant due to its higher solubility, and long SLG (L-SLG) was precipitated due to its lower solubilization in the aqueous ethanol solution. Furthermore, large and aggregated particles (200–300 nm) were formed once the SLG concentration further rose to 10%. However, in the study of Qin et al. [87], who investigated debranched starch nanoparticles as seen by transmission electron microscopy (TEM), (DBS-NPs) produced at 25, 60, and 90 °C had DPs with various average levels of polymerization. Since the DPs of DBS were greater than 10 (average DP = 13–24), TEM was used to examine the morphology and size distribution of DBS-NPs. The DBS generated spherical NPs (40–200 nm) with some aggregation. Ogawa and Putaux [88] reported that traditional TEM imaging provides two-dimensional pictures, making it impossible to access height information perpendicular to the observation plane.

#### 5.3.3. Scanning Electron Microscopy

Scanning electron microscopy is a typical technique used for observing a material’s microstructure and morphology (SEM). SEM modalities for material characterization (including biomaterials) include X-ray mapping, secondary imaging of electrons, backscattered imaging of electrons and electron channels, and Auger electron microscopy. SEM may be used to analyze and interpret micron- or nanometer-scale data [89]. An electron microscope’s resolution for field scanning can be as low as 1 nm. Because of its great field depth, SEM also allows for three-dimensional observation and analysis of samples. The more sample information that is supplied, the larger the depth of field [2]. Liu et al. [66] used SEM in waxy corn nanocomposite films. The films were promptly shattered after being frozen in liquid nitrogen in order to observe the cross-section. The fracture cross-sections and surfaces were gold-sputtered before being photographed. This resulted in SNPs that ranged in size from 20 to 30 nm. The surface of the corn starch films without SNPs was smooth, but as the SNP content rose, the nanocomposite films began to display some surface roughness. Meanwhile, Gonde et al. [90] reported on rice starch nanoparticles and their inherent surface morphology and found that the native rice starch granules were mostly polyhedral in shape and had a smooth surface. SNPs generated by acoustic and hydrodynamic cavitation were needle-shaped.

#### 5.3.4. Fourier Transform Infrared Spectroscopy

Fourier transform infrared (FTIR) spectroscopy provides helpful information on the structure’s functional groups by studying the vibrant frequencies of the chemical bonds. The molecules’ vibratory activation intensity is between 1013 and 1014 Hz, which corresponds to infrared light. This enables IR spectroscopy to examine and perform quantitative and qualitative investigations on self-assembled functional groups arranged on nano-paragon surfaces [91]. The benefit of FTIR is that it allows users to examine the layers and overlapping phases of nanoparticles on the ATR element. This technique’s molecular data collection assists users in determining the structural and conformational changes in self-assembled functional coordinating groups on nanoparticle surfaces [92]. Tao et al. [93] investigated, via an FTIR spectrophotometer, Capsaicin (CAP) and indica rice starch nanoparticles (IRSNPs) combined with bone dry potassium bromide (KBr) powders at a ratio of 1:100. FTIR spectra were collected with a resolution of 2 cm^−1^ between 4000 and 400 cm^−1^. The FTIR spectrum of CAP-IRSNPs demonstrated that CAP and IRSNPs successfully interacted. In comparison to free CAP, the distinctive peak of CAP-IRSNPs was pushed to a lower wavenumber, indicating a strong intermolecular hydrogen bond.

### 5.4. Structural Properties

#### 5.4.1. X-ray Diffraction

X-ray diffraction (XRD) is a useful technique for studying nano-materials (materials with structural properties in a range between 1 and 100 nm with at least one size). The intensities obtained using XRD can give quantitative and precise information on nuclear structures at interfaces [94]. Nano-materials have a suitable microstructure length relative to the critical length scales of physical processes, which gives them unique mechanical, optical, and electrical characteristics. From phase composition to crystallite size, and from lattice strain to crystallographic orientation, nano-material XRD provides a plethora of information [85]. Kim et al. [95] used XRD With the use of an X-diffractometer and radiation given at a specific voltage and current of 40 kV and 30 mA, respectively. X-ray diffractograms of native maize starches and starch hydrolysates were obtained, and they showed the characteristic of native starches A, B, and C crystalline structural patterns. The A-type crystalline structure is represented by diffraction peaks at Bragg angles (2 u) of 158, 178, 188, and 238 in the case of Wx and normal maize starches. For high AM maize and potato starches, a sharp peak emerged at 2 u 178, and B-type crystals were detected at 2 u 5.68, 158, 208, 228, and 242. No and Kampus [96] analyzed the structure of mangrove starch at 40 kV and a current of 30 mA of electricity using Cu Kα radiation at a 1.5418 A wavelength and scanned from 0. 0050 (2Ɵ/s), which indicated the beam gathers in the detector, allowing the production of the diffractogram. Amorphous polymers have a large peak on the diffractogram while X-ray crystalline polymers have a sharp peak. With two diffraction peaks at 2 = 15.3 and 22.5 and a double peak close to 17.5, the XRD pattern of DSNPs-60 was consistent with that of DBS, indicating that both substances possessed A-type crystalline structures. Debranched starch and ethanol appeared to form a V-type complex, as demonstrated by the production of a crystalline structure in the form of a V [37].

#### 5.4.2. Nuclear Magnetic Resonance Spectroscopy Analysis

Nuclear magnetic resonance (NMR) is a strong non-destructive analytical technique that uses magnetic field exposure on excited nuclei. It is an electrical phenomenon caused by the unique magnetic properties of particular atomic nuclei [97]. NMR is also often utilized in advanced imaging technologies such as magnetic resonance imaging (MRI). It offers extensive information on the molecular structure, dynamics, response status, and chemical environment by detecting the return of nuclei to their base level of energy [98]. Flanagan et al. [99] analyzed the molecular properties of lotus seed starch nanoparticles by using that at 100.62 MHz, and the resonance probe was a 7 mm H/X CP-MAS probe, proving that lotus seed starch nanoparticles produced through enzymatic hydrolysis and those produced with ultrasonic-assisted enzymatic hydrolysis both display a bimodal structure consisting of two glucose residues in the C1 region, indicating that they are type B starches. As a result, the strength of the double-helix structure in the starch crystal region may be indicated in the signal peak in the C1 region. The strength of the double helix structure increases with increased signal peak intensity. The lotus seed starch nano-particles produced through enzymatic hydrolysis have a stronger double helix structure than those produced via ultrasonic-assisted enzymatic hydrolysis. Patel and Chudasama [100] analyzed the chemical kinetics using this method, which is another use. So, we can also check the solvents or other materials’ other properties using this method, as well as the dielectric static polarity. NMR’s limitations include poor availability of large atomic weights, but it is great for more precisely determining the structure. It gives specific structural detail, is moderately sensitive, and is quantitative (with the proper experimental setup), allowing concentrations to be estimated. Ben-Tal et al. [101] reported that, generally, solution-phase NMR spectroscopy meets most or all of the aforementioned criteria, allowing concentrations to be estimated. It is also suitable for a wide range of nuclei prevalent in organic and organometallic processes.

#### 5.4.3. Dynamic Light Scattering

The size distribution of tiny particles in suspension and/or polymers in a solution can be determined using dynamic light scattering (DLS), a physics method. To investigate temporal changes in DLS, the amplitude or photon self-correlation technique (also known as photon correlation spectroscopy or almost-elastic light scattering) is commonly utilized. Fluids, such as condensed polymer solutions, are also tested using DLS [102,103]. This device detects light reflected from a sample-carrying flow stream in a cylindrical capillary that is powered by a laser beam that is focused in the opposite direction. The sample-carrying flow stream and the laser beam in the system are coaxial, and the laser beam and the detectors are co-planar. The ability to measure scattering at lesser degrees is significantly impacted by the circular shape (accessible in a 0 to 180° range). DLS concurrently measures the scattering at a unique set of scattering angles using a variety of detectors that act as a base for the simultaneous dLS detecting concept and systems for scattering commercial light [104]. Pyttlik et al. [105] clarified that hydrodynamic particle sizes between nanometers and micrometers can be measured using dynamic light scattering. The relevant specific mean residence times can be resolved with a 1 s integration time. Microseconds and nanoseconds have different lengths. Ali and Gad [106] used dLS to measure the size of starch nanoparticles (StNPs). Prior to analysis, all mixtures were filtered using disposable 0.45 m pore filters with a concentration of 1.0 mg/mL and the given size was approximately 50 to 250 nm, with this becoming the mean size. In the aqueous state, nanoparticles have a tendency to accumulate and can provide clustered particles during DLS measurements as opposed to single particle sizes.

## 6. Applications of Starch Nano-Particles (SNPs)

### 6.1. SNPs in Non-Food Application

#### 6.1.1. Carrier in Drug Delivery

The use of polymeric nanoparticles in drug delivery systems has attracted much interest because of their unique properties. The medium on which the medication is loaded, entrapped, encapsulated, or bonded is known as the nanoparticle matrix [107,108]. SNPs have a number of features that make them a viable option for use as a drug carrier, including non-toxicity, decomposability, and the ability to transport medicines. Santander-Ortega and colleagues examined the use of starch nanoparticles as a transdermal drug delivery system (TDDS) in 2010. The difficulty with such a system’s medication delivery was due to the skin’s barrier characteristics, which reduced the effective drug distribution and thus needed to be addressed. Nano drug delivery methods provide a number of advantages, including improved encapsulation efficiency, excellent biochemical or enzymatic degradation resistance, and the unique ability to generate a controlled release [109]. Because of their enhanced hydrophilicity, biocompatibility, and biodegradability, their use is gaining a great deal of attention. Modified starch nanoparticles were utilized as indomethacin carriers [110]. Xiao et al. [111] used a potentially potent technique to sustain the release of therapeutic nanoparticles in a way that is particular to a given cell, tissue, or disease. Diadehyde starch nanoparticles (DASNP), a novel drug carrier, can sustain the loading and release of the anticancer medication 5-fluorouracil (5-Fu). It was discovered that modified starch nanoparticles exhibited enhanced loading capacity and in vitro release behavior for indomethacin medication [109,110]. Diadehyde starch nanoparticles (DASNP), a novel drug carrier, can sustain the loading and release of the anticancer medication 5-fluorouracil (5-Fu). These outcomes showed that the DASNP created in this study has the potential to be a strong medication carrier. El-Feky et al. [112] reported that native starch nanoparticles containing insoluble medications such as indomethacin (IND) and acyclovir were created using a green nanoprecipitation method (ACV). These factors should be taken into consideration when creating various formulations of crosslinked starch nanoparticles (TPP-StNPs) loaded with insoluble medicines. Entrapment effectiveness and in vitro release were used to assess the sustained release of medicines. Xie et al. [113] reported that, due to the lower crystallinity of anionic and cationic groups, amphoteric cassava starch nanoparticles (CA-CANPs) encourage the drug to be loaded inside nanoparticles, improving the drug loading capacity and maintaining slow-release performance in a typical body fluid environment. Troncoso and Torres [114] clarified that, due to their hydrophobic characteristics, acetylated and hydroxypropylated starches are also among the most beneficial starches for drug-delivery applications (Figure 7). When functional acetylated groups (CH_3_CO) are added, the free hydroxyl groups in the branched chains of the starch polymers react to form a particular ester. When starch is esterified, reactive reagents including anhydrous acetic acid, vinyl acetate, and octenyl succinic anhydride (OSA) are used in the presence of an alkaline catalyst (NaOH, KOH, Ca(OH)_2_, Na_2_CO_3_).

#### 6.1.2. Nano Starch in Food Packaging Industry

Nanotechnology’s critical role in food packaging is regarded as the greatest commercial use in the food industry [115]. In recent years, there has been an increased focus on food packaging materials research and innovation, ranging from films to carbon nanotubes to waxy nano-coatings for specific foods. Nanoparticles might aid in the development of novel food packaging materials with improved characteristics to extend the shelf life [116]. Apart from their antibacterial properties, nanoparticles can be utilized as a transporter of antioxidants, enzymes, tastes, anti-browning agents, and other compounds to increase the shelf life even after they have been opened [117,118,119]. Boumans claimed that a starch-based colloidal coating filled with antibacterial nanoparticles provided protection and food packaging. Water vapor permeability and the oxygen transfer rate are the two most important barrier characteristics in packaging materials. The addition of 30–40% waxy maize SNPs to sorbitol-pullulan films resulted in a substantial reduction in water vapor permeability [120,121]. When potato SNPs were introduced at varying levels to pea starch films, their physicochemical characteristics improved. Films with SNPs showed better tensile strength and less water vapor permeability [122,123,124]. Roy et al. [125] prepared nano starch combined with native starch at various concentrations, i.e., 0.5, 1, 2, 5, and 10%, to produce nano starch-based composite films. Starch and nano starch (5%, dry basis, *w*/*w*), glycerol (2.5%, *w*/*w*), and oil (2 g/L) were used to make composite films. Gelatinized suspensions (20 mL) were then placed into Petri plates just after film-forming dispersion had been heated in a microwave (2350 W) for 30 to 60 s while being mixed continuously. The film-forming solution-coated plates were then dried for 8 to 10 h at 50 °C in an air-circulating oven. Then, the Petri plates’ dry films were peeled away, which proved that the composite film degraded more quickly, proving the starch nanoparticles improved the biodegradability of the films. The SNPs had an impact on the composites’ ability to degrade more quickly because, in soil, water diffuses into the polymer sample, inflating it and accelerating decomposition through an increase in microbial growth. Chavan et al. [126] reported that nano-size, which expedites the water vapor resistance, causes the nano starch to increase the density of nano-composite coatings. They are not a suitable replacement for petroleum-based plastic films in all situations involving food packaging.

#### 6.1.3. Adsorbents in Water Treatment

Adsorption is thought to be the most promising strategy for treating wastewater, with its straightforward operation process, low cost, and reduced sludge formation. Owing to their unique qualities, such as high adsorption strength, greater surface area, and chemical stability, nanomaterial adsorbents have gained considerable attention in this area [127]. After chemical modification, starch nanoparticles can be used as adsorbents to remove aromatic and organic pollutants from water. Activated carbon has long been used to clean wastewater, but its application is limited due to its expensive cost. The use of synthetic polymers may be a solution to this problem. Over the last decade, the renewable and cost-effective alternative has received a great deal of attention. Biopolymers have been discovered to be the best choice for this application. Polystyrene-modified cellulose nano whiskers have recently been investigated for the removal of aromatic organic chemicals from wastewater. After altering it by grafting with stearate, Alila et al. [128] were the first to disclose the possible use of SNPs as adsorbents. When compared to their native counterparts, nanoparticles made from starch have shown higher efficiency and capability due to their increased surface area. Chemical manipulation of SNPs (grafting, crosslinking, etc.) has improved adsorption capacity efficiency even more. When 2-naphthol and nitrobenzene were employed, the adsorption capacity of SNPs grafted with stearate increased from 150 to 900 mol g^−1^ compared to 50 and 40 mol g^−1^ when unmodified starch was used [128,129,130,131]. The major driving force behind the adsorption is thought to be Van der Waals interactions between the grafted chain and organic molecules. The planar structure of aromatic organic solutes further facilitates molecular interaction inside the domain created by grafted SNPs. It is reported that the regeneration and distribution of pollutants in line with the equilibrium circumstances will determine the adsorption process, as well as the adsorption coefficient. Additionally, the beginning of the transformation of the ion structure is promoted by a redox reaction with persistent inorganic contaminants [132,133]. Gad et al. [134] investigated the anionic StNP citrate as an adsorbent for crystal violet dye removal from a watery arrangement. The findings of the adsorption procedure showed that at pH 8.5 and a temperature of 50 °C for 120 min, the highest color evacuation is obtained. Figure 8 shows SNPs adsorbents in water treatment.

#### 6.1.4. Emulsion Stabilizer

Food processing and preservation need a variety of emulsion stabilizers. As emulsion stabilizers in meals, different particles such as hydrophobic silica, carbon nanotubes, latex, bacterial cellulose nano-crystals, and micro-fibrillated cellulose have been employed. SNPs have gained popularity as emulsion stabilizers in recent years. It was discovered that combining a small quantity of an SNP suspension with water and paraffin produced a stable emulsion. The suspension was stabilized for more than two months after SNP was added at a concentration of more than 0.02 percent by weight. However, phase separation occurred when the suspension was heated for 2 h at 80 °C [135]. In addition to the food industry, SNPs can be utilized as emulsion stabilizers in cosmetics and medicines.

Haaj et al. [136] reported the use of persulfate as an initiator and starch nanoparticles (SNPs) as the only stabilizer. Without using any chemical reagents, high-power ultrasonication was used to produce the SNPs in water. The gradually decreasing particle sizes of the polymer solution as the SNP loading increased supported the importance of the SNPs in the stabilizing process. It was demonstrated that nanocomposites made using the Pickering emulsion method have higher optical transparency than those made using the ex situ mixing technique. Ko and Kim [137] confirmed that the SNP-stabilized emulsion remained stable in an acidic medium. The use of SNPs as an emulsifying agent may be restricted by their limits in stabilizing emulsions. Kamwilaisak et al. [138] investigated the production of SNPs by acid hydrolysis and their use as an emulsion stabilizer, indicating that the Pickering emulsion method showed pseudoplastic fluid behavior and shear thinning features.

## 7. Conclusions

Because of their enormous potential as nano-carriers in many bio applications such as medicine, cosmetics, and food, nanoparticles have recently seen a significant increase in terms of scientific research. SNPs have a distinct advantage over other types of nanoparticles owing to their excellent biocompatibility and flexibility, since there are a variety of possible synthesis techniques that allow for the customization of final shape characteristics such as size and size distribution. SNPs were created utilizing a variety of synthesis techniques using starch from various sources. In most cases, the manufacturing technique, as well as the kind of starch used, will influence the final characteristics of the nano-particles. Maize starch is one of the most widely utilized types of starch for SNP synthesis, either alone or in combination with other starch such as potato, pea, or faba bean, and has been proven to result in tiny particle sizes. SNPs with small sizes may be obtained using both nano-precipitation and ultrasonic techniques.

## Figures and Tables

**Figure 1 molecules-27-05497-f001:**
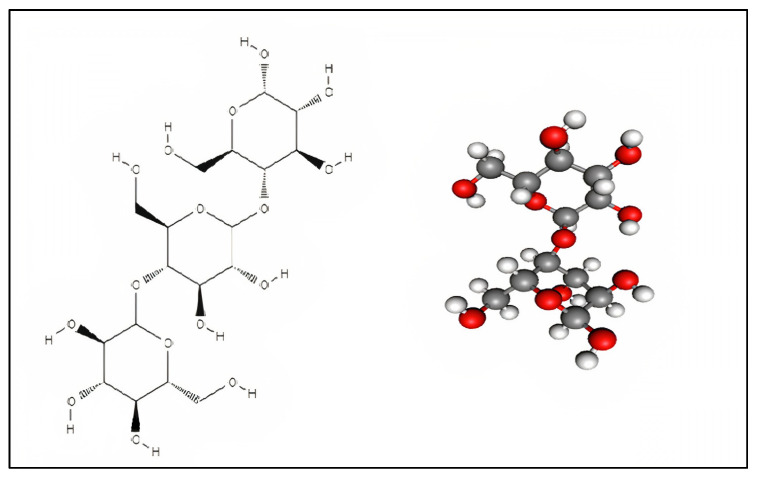
Basic structure of the starch molecule.

**Figure 2 molecules-27-05497-f002:**
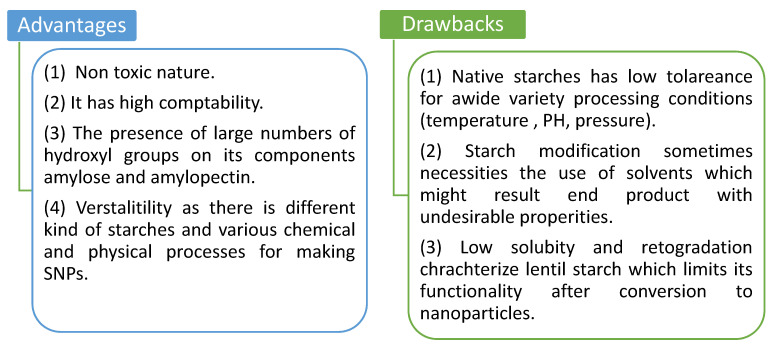
Advantages and drawbacks for preparation of starch as nanoparticles (SNPs).

**Figure 3 molecules-27-05497-f003:**
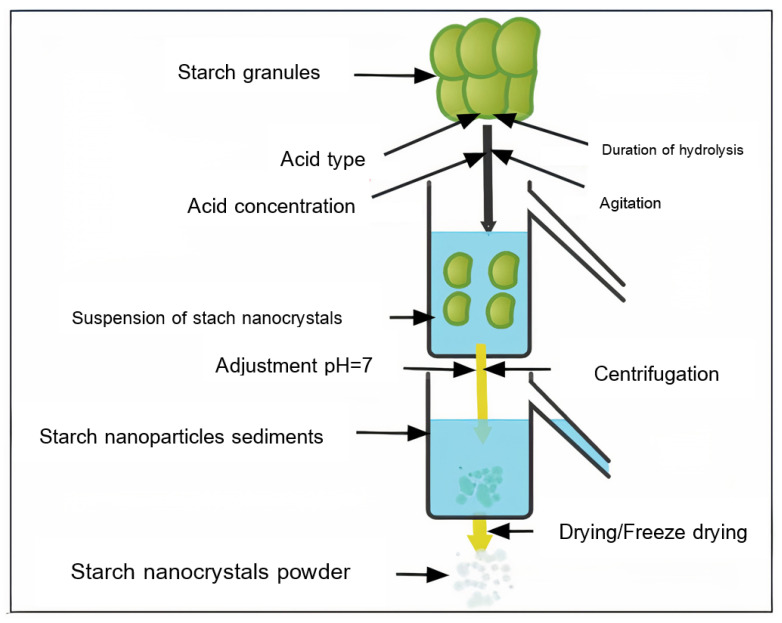
Acid hydrolysis for preparation of starch nanoparticles.

**Figure 4 molecules-27-05497-f004:**
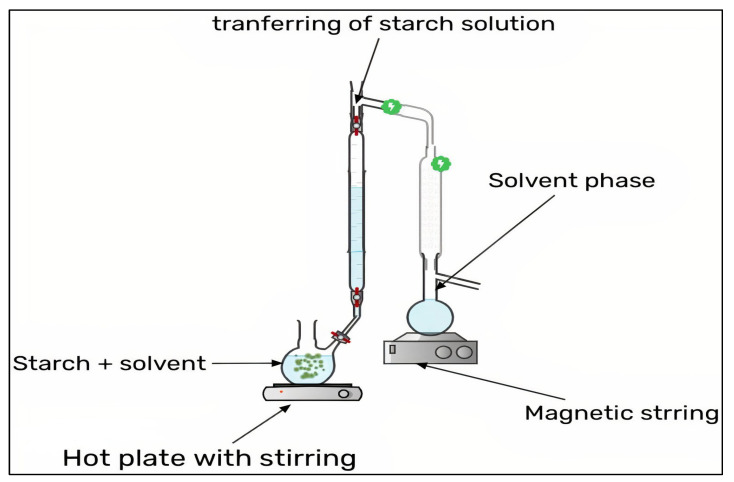
Nano-precipitation method of SNPs.

**Figure 5 molecules-27-05497-f005:**
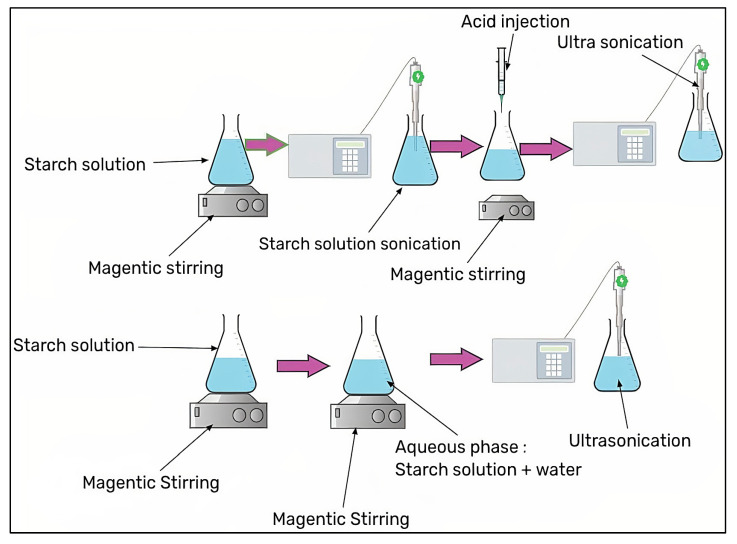
Ultrasonication with/without acid hydrolysis.

**Figure 6 molecules-27-05497-f006:**
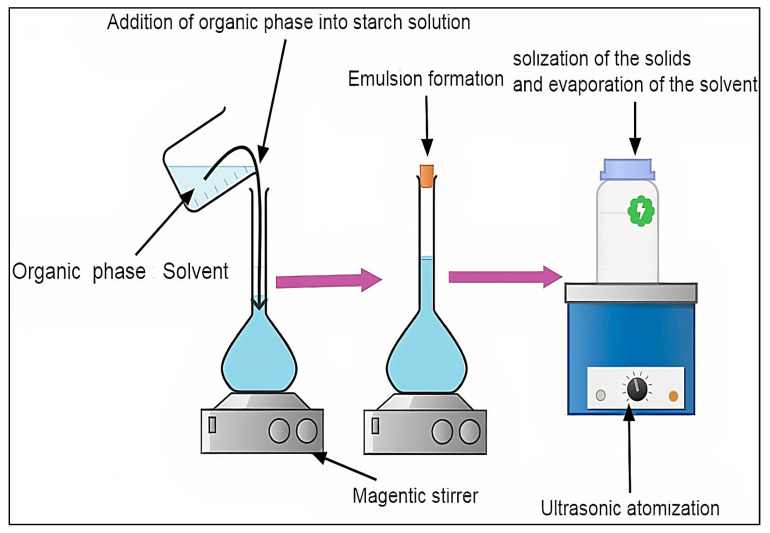
Ultrasonic atomization method.

**Figure 7 molecules-27-05497-f007:**
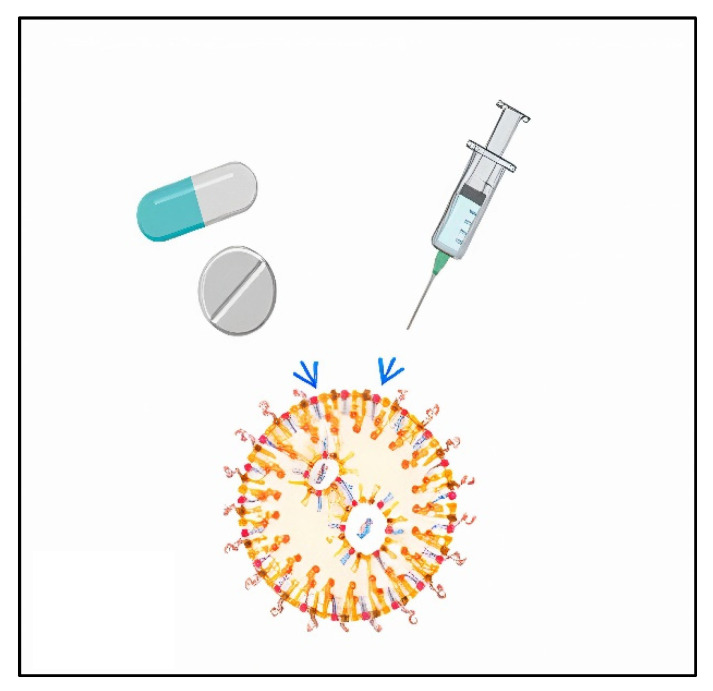
SNPs carrier as drug delivery.

**Figure 8 molecules-27-05497-f008:**
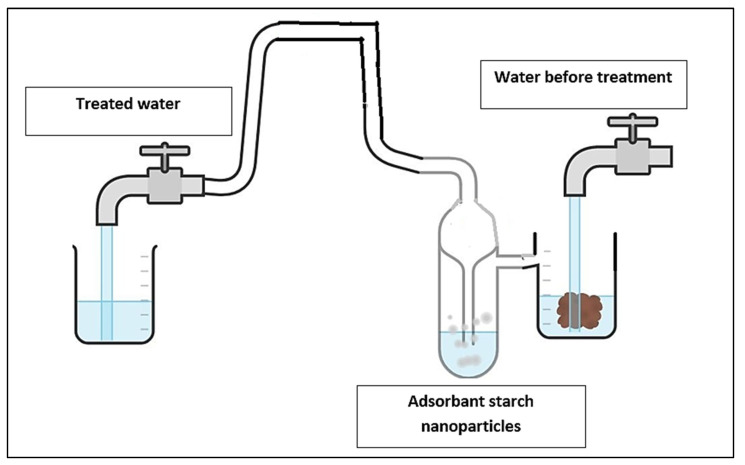
SNPs adsorbents in water treatment.

**Table 1 molecules-27-05497-t001:** Production rates of starch nanoparticle preparation methods.

Preparation Methods	Sources	Morphology	Size (nm)	Yield (%)	References
Ultrasonication	Maize starch	platelet or granular	30 to 100	≈100	[57]
Acid hydrolysis and ultrasound	Maize starch	Globular shapes	50 to 90	78	[58]
Nanoprecipitation	Maize starch	Spherical	135 to 155	ND	[60]
Acid hydrolysis	Corn starch	Nanoplatelets	107	ND	[55]
Milling	Maize starch	Gel-like	245	ND	[59]
Enzymolysis	Maize starch	Irregular	2.4 to 6.7	29.8	[56]
Ultrasonication	Maize starch	platelet	40 nm	-	[62]
Homogenization	Maize starch	Smaller starch Granules, pores	540	ND	[40]
Enzymolysis	Maize starch	Spherical	162 ± 23, 301	18	[63]
Acid hydrolysis and ultrasound	Maize starch	spherical and ellipsoidal	20–250		[64]

## Data Availability

All data generated or analyzed during this study are included in this article.

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
