# Peer review of "Recent Trends in the Preparation of Nano-Starch Particles"

_molecules, 2022, doi:10.3390/molecules27175497_

Round 1

Reviewer 1 Report

The research is aimed to address the main manufacturing process, characterization of properties for starch nanoparticles and the emerging applications for these novel biomaterials. SNPs have a distinct advantage over other types of nanoparticles owing to their excellent biocompatibility and flexibility. In most cases, the manufacturing technique as well as the kind of starch used will influence the final characteristics of the nano-particles. Because of its enormous potential as nano-carriers in many bio applications such as medicine, cosmetics, and food, the study of nanoparticles has seen a significant increase in terms of scientific research. It would seem to be a well-defined topic and of some importance. However, the article also has many errors that need to be modified:

1.  Line 109The classification basis of the preparation method of starch nanoparticles would be better.

2.   Line 118-441The preparation methods of starch nanoparticles are divided into three categories in the previous section. Why are they described according to conventional and non-conventional classification in the following section?

3. Line 443The advantages and disadvantages of the preparation method of starch nanoparticles are relatively messy, and the table can be attached to make a clear contrast.

4.  The problem of the whole article is that the comprehensive refining is not enough, the literature is piled up too much, and the generalization and summarization of paragraphs needs to be increased.

5.  Picture pixels are not enough, the clarity of these pictures needs to be increased, and some places have background color.

Author Response

Reviewer 1

However, the article also has many errors that need to be modified:

1.Line 109:The classification basis of the preparation method of starch nanoparticles would be better.

Reply: done

2.Line 118-441:The preparation methods of starch nano-particlesare divided into three categories in the previous section. Why are they described according to conventional and non-conventional classification in the following section?

Reply: done we removed the second classification.

3.Line 443:The advantages and disadvantages of the preparation method of starch nanoparticles are relatively messy, and the table can be attached to make a clear contrast.

Reply: done see Table 1

4.The problem of the whole article is that the comprehensive refining is not enough, the literature is piled up too much, and the generalization and summarization of paragraphs needs to be increased.

Reply: done, we have made a good revision and summarization through the manuscript.

5.Picture pixels are not enough, the clarity of these pictures needs to be increased, and some places have background color.

Reply: done we have increased the clarity of all pictures.

Reviewer 2 Report

The article is interesting and cover many approaches to the starch nanoparticle

Some points must be revised 

The abstract need carefully revision

The low digestibility of starch in humans is not applicable

The title 6.1.2. it should be nanostarch not nanocomposite

The applications of nanostarch need more details

Some references may be helpful for authors

Ecofriendly synthesis of biosynthesized copper nanoparticles with starch-based nanocomposite: antimicrobial, antioxidant, and anticancer activities

Immobilization of L-methionine γ-lyase on different cellulosic materials and its potential application in green-selective synthesis of volatile sulfur compounds

Simple, economic, ecofriendly method to extract starch nanoparticles from potato peel waste for biological applications

Author Response

Reviewer 2

The article is interesting and cover many approaches to the starchnanoparticle

Some points must be revised

The abstract need carefully revision

Reply: done

The low digestibility of starch in humans is not applicable

Reply: done we have modified this sentence to be “Starch is affected by several limitations e.g. retro-gradation, high viscosity even at low concentrations, handling issues, poor freeze-thaw stability, low process tolerance, and gel opacity” in the abstract section.

The title 6.1.2. it should be nanostarch not nanocomposite

Reply: done

The applications of nanostarch need more details

Reply: done

Some references may be helpful for authors

Ecofriendly synthesis of biosynthesized copper nanoparticles withstarch-based nanocomposite: antimicrobial, antioxidant, andanticancer activities

Immobilization of L-methionine γ-lyase on different cellulosicmaterials and its potential application in green-selective synthesisof volatile sulfur compounds

Simple, economic, ecofriendly method to extract starchnanoparticles from potato peel waste for biological applications

Reply: Thanks for your help.

Round 2

Reviewer 1 Report

This article was improved compared with previous manuscript. I believed it has met the request of “molecules”, and could be accepted.

Reviewer 2 Report

Accept in present form